# Efficacy of rupatadine in reducing the incidence of dengue haemorrhagic fever in patients with acute dengue: A randomised, double blind, placebo-controlled trial

**Gathsaurie Neelika Malavige** [1,2]* , **Chandima Jeewandara**[1] , **Ananda Wijewickrama**[3] , **Dumni Gunasinghe**[1] , **Sameera D. Mahapatuna**[1] , **Chathurika Gangani**[1] , **Vimalahan Vimalachandran**[1] , **Geethal Jayarathna**[1] , **Yashoda Perera**[1] , **Chandanie Wanigatunga**[1] , **Harsha Dissanayake**[1] , **Shamini Prathapan**[1] , **Eranga Narangoda**[3] , **Damayanthi Idampitiya**[3] , **Laksiri Gomes**[1] , **Samurdhi Wickramanayake**[1] , **Pramodth Sahabandu**[1] , **Graham S. Ogg**[1,2]

1 Faculty of Medical Sciences, University of Sri Jayewardenepura, Nugegoda, Sri Lanka, 2 MRC Human Immunology Unit, MRC Weatherall Institute of Molecular Medicine, University of Oxford, Oxford, United Kingdom, 3 National Institute of Infectious Diseases, Angoda, Sri Lanka

☉ These authors contributed equally to this work.
* gathsaurie.malavige@ndm.ox.ac.uk

## Abstract

### Background

Rupatadine was previously shown to reduce endothelial dysfunction in vitro, reduced vascular leak in dengue mouse models and to reduce the extent of pleural effusions and thrombocytopenia in patients with acute dengue. Therefore, we sought to determine the efficacy of rupatadine in reducing the incidence of dengue haemorrhagic fever (DHF) in patients with acute dengue.

### Methods and findings

A phase 2, randomised, double blind, placebo controlled clinical trial was carried out in patients with acute dengue in Sri Lanka in an outpatient setting. Patients with ≤3 days since the onset of illness were either recruited to the treatment arm of oral rupatadine 40mg for 5 days (n = 123) or the placebo arm (n = 126). Clinical and laboratory features were measured daily to assess development of DHF and other complications. 12 (9.7%) patients developed DHF in the treatment arm compared to 22 (17.5%) who were on the placebo although this was not significant (p = 0.09, relative risk 0.68, 95% CI 0.41 to 1.08). Rupatadine also significantly reduced (p = 0.01) the proportion of patients with platelet counts <50,000 cells/mm$^3$ and significantly reduced (p = 0.04) persisting vomiting, headache and hepatic tenderness (p<0.0001) in patients. There was a significant difference in the duration of illness (p = 0.0002) although the proportion of individuals who required hospital admission in both treatment arms. Only 2 patients on rupatadine and 3 patients on the placebo developed shock, while bleeding manifestations were seen in 6 patients on rupatadine and 7 patients on the placebo.

**Data Availability Statement:** All relevant data are within the manuscript and its Supporting Information files.

**Funding:** The funding was provided by the Centre for Dengue Research, University of Sri Jayewardenepura, Sri Lanka (GNM) and the MRC, UK and NIHR Biomedical Research Centre for the funding (GSO). The funders had no role in study design, data collection and analysis, decision to publish, or preparation of the manuscript.

**Competing interests:** The authors have declared that no competing interests exist.

## Conclusions

Rupatadine appeared to be safe and well tolerated and showed a trend towards a reducing proportion of patients with acute dengue who developed DHF. Its usefulness when used in combination with other treatment modalities should be explored.

## Trial registration

International Clinical Trials Registration Platform: SLCTR/2017/024.

### Author summary

Rupatadine was previously shown to reduce endothelial dysfunction in vitro, reduced vascular leak in dengue mouse models and to reduce the extent of pleural effusions and thrombocytopenia in a post-hoc analysis, in patients with acute dengue. Therefore, we sought to determine the efficacy of rupatadine in reducing the incidence of dengue haemorrhagic fever (DHF) in patients with acute dengue. A phase 2, randomised, double blind, placebo controlled clinical trial was carried out in patients with acute dengue in Sri Lanka in an outpatient setting. Patients with ≤3 days since the onset of illness were either recruited to the treatment arm of oral rupatadine 40mg for 5 days (n = 123) or the placebo arm (n = 126). We found that patients given rupatadine were less likely to develop DHF than those on the placebo (relative risk 0.68, 95% CI 0.41 to 1.08) although not significant. Rupatadine also significantly reduced (p = 0.01) the proportion of patients with platelet counts <50,000 cells/mm$^3$ and significantly reduced (p = 0.04) persisting vomiting, headache and hepatic tenderness (p<0.0001) in patients. Rupatadine appeared to be safe and well tolerated and showed a trend towards a reduced proportion of patients with acute dengue who developed DHF.

## Introduction

Dengue infections are one of the most rapidly emerging vector-borne viral infections, with the number of global infections increasing from 23 million in 1990 to 104 million in 2017 [1]. The global age-standardised death rate increased in parallel from 0.31 per 100,000 population in 1990 to 0.53 per 100,000 population in 2017 [1]. Multiple factors are thought to play a role in this increased burden, which include global warming due to climate change, increased mobility, and urbanization [2,3]. 70% of the dengue infections occur in Asia, and dengue is a major public health problem in Sri Lanka and many other resource poor countries [4,5]. There are currently no specific treatments for dengue; and intense monitoring to detect complications and administration of fluids are the mainstay of dengue management [6].

Although the majority of infections with the dengue virus (DENV), result in asymptomatic or mild infection, some individuals develop complications such as dengue haemorrhagic fever (DHF) and organ dysfunction. Endothelial dysfunction leading to vascular leak is the main cause of DHF [7]. A dysfunctional immune response to the DENV results in the production of many inflammatory cytokines such as IL-1β and TNFα, inflammatory lipid mediators such as platelet activating factor (PAF), and the dengue NS1 protein, which are all thought to contribute to the vascular leak [8–10]. Previously we showed that PAF caused endothelial dysfunction in a dose dependent manner. PAF receptor blockade significantly increased the ZO-1

expression and increased the endothelial electrical resistance that was reduced by sera of patients with dengue shock syndrome [11]. A preliminary clinical trial carried out by us showed that rupatadine was safe and well tolerated in patients with acute dengue and significantly reduced the extent of fluid leakage and reduction in platelet counts [12]. Furthermore, we found that in when given early (≤ 3 days since onset of illness), rupatadine appeared to reduce the proportion of individuals developing ascites and pleural effusions, although this was not significant, as the study was not adequately powered to assess this [12].

While the health care facilities in many countries are overwhelmed due to the ongoing COVID-19 pandemic, many countries in Asia are further affected due to the double burden of dengue and COVID-19 [13,14]. Both dengue and COVID-19 can have a somewhat similar clinical presentation with fever, myalgia, headaches, loss of appetite and especially in children sore throat [15,16]. While most patients who develop severe clinical disease due to COVID-19, usually do so after 6 to 7 days of illness, patients with acute dengue can suddenly develop fluid leakage and develop DHF [6,16]. Therefore, there is an urgent need to find therapeutics to reduce the incidence of DHF in patients with acute dengue. Here we describe a phase 2, randomised, double blind, placebo-controlled trial, evaluating the efficacy of rupatadine in reducing the incidence of DHF.

## Methods

### Ethics statement

The trial was approved by the Ethics Review Committee of the University of Sri Jayewardenepura, Sri Lanka (37/17) and also the SCOCT of the Ministry of Health, Sri Lanka. The trial was registered at the Sri Lanka Clinical trial registry on the 21st of July 2017 (International Clinical Trials Registration Platform: SLCTR/2017/024) and the trial recruited patients from December 2017 to March 2020. All methods involving human patients were performed in accordance with the relevant guidelines and regulations. All patients gave informed written consent.

### Trial design and oversight

The trial was an investigator-led phase II, randomized, double blind, placebo-controlled trial, consisting of two arms, which were oral rupatadine 40mg and the placebo for a duration of 5 days. This study was carried out at the Outpatient Department (OPD) of the National Institute of Infectious Diseases, which is a tertiary care hospital in Colombo District, Sri Lanka in 2017 December to 2020 March.

An interim analysis was carried out after recruiting half of the number of patients (n = 140) in order to find out if there were any safety concerns. Details regarding the trial protocol are available as S1 Appendix. We adhered to the CONSORT guidelines for publication when analysing and reporting our data [17]. An independent drug safety monitoring board (DSMB), which consisted of an experienced biostatistician, a clinical pharmacologist and an Infectious Diseases specialist and a senior General Practitioner assessed the safety outcomes and the overall study integrity.

### Patient recruitment and follow-up

Patients with a suspected dengue infection who presented to the OPD of the National Institute of Infectious Diseases, with a febrile illness of ≤ 3 days duration, who were tested positive for the dengue point of care, NS1 antigen test (SD Bioline, South Korea), were recruited following informed written consent. Dengue was confirmed by detection of virus by quantitative multiplex, real time PCR [18] as previously described. All pregnant women, those who report

reactions to antihistamines or relevant excipients, those who are alcohol dependent or abuse drugs or those with previously diagnosed hepatic or renal impairment were excluded.

At the initial and subsequent visits, which was for 5 days, the patients were assessed for the presence of any warning signs and clinical parameters were recorded by medical personnel at the outpatient department. If the patients fulfilled any criteria for hospital admission as per National Dengue Management Guidelines in Sri Lanka, or if they wished to seek admission to hospital (due to personal reasons), they were assessed several times each day in hospital as per National Guidelines [19]. The medical personnel managing the patients in the OPD and in the wards, recorded the clinical data, and decided on fluid management and when to discharge from hospital and had no knowledge of which drug the patients were given. In the patients who were not admitted, the clinical features and the blood counts were assessed daily. If the patients were admitted, their blood counts were assessed several times a day and ultrasound scans were done daily. If presence of fluid was detected along with the presence of thrombocytopenia, they were classified as having DHF has per WHO 2011 disease classification [6]. The investigators who were involved in recruiting and handing the drug envelopes, recorded all clinical data from the hospital records. Details of clinical disease severity such as development of DHF, bleeding and acute liver failure and adverse events were recorded and reported to the Ethics Review Committee. Serious adverse events were reported to the Ethics Review Committee and to the independent DSMB.

## Randomization

The randomization for rupatadine 40mg or the placebo was done in 1:1 ratio using a computerized random number generator of sequential patients with dengue attending the OPD at the hospital. Once the patient was recruited, he/she was given a unique code. The patients who received the 40mg dose received 4 tablets of 10mg rupatadine tablets daily for 5 days from the day of recruitment and the control group, 4 placebo tablets daily for 5 days. The drug was given for 5 days as patients with acute dengue are unlikely to become acutely ill for more than 5 days from the day of recruitment ($\leq$ 3 days since onset of illness), unless they develop significant complications.

The patients received the drugs once the randomization was completed and they were assigned to a study arm and the drugs were administered each morning. The rupatadine tablets were provided by the pharmaceutical manufacturers Dr. Reddy in India, and Sri Lanka Pharmaceutical Manufacturing Corporation provided the placebo which was the same size, shape and colour. They played no part in the study design, analysis or reporting. Participants and investigators were kept blind to the treatment allocation for the duration of study. All groups received the standard supportive care treatment as per national guidelines with no other differences between groups [19].

## Trial end points

The primary objective in this study, was to evaluate the reduction in the proportion of patients developing DHF when given rupatadine 40mg/day for 5 days in acute dengue infection. The secondary objectives were to assess reduction in those that developing liver failure, shock, the need of the use of colloids and blood transfusions. Details of definitions and measurements used detection of fluid leakage, liver failure, shock and duration of illness are given in Supplementary appendix 1. The primary and secondary outcomes were analysed by assessing the differences in the proportion of individuals on rupatadine vs placebo who developed DHF, liver failure, shock, who required colloids, required blood transfusions and the differences in the duration of acute illness. The first day of illness was defined as the day in which the patient

developed fever and the day of recovery was defined as patient being afebrile for 24 hours, the platelet counts rising to >50,000 cells/mm$^3$ or a rise of 20% from the lowest platelet value, and the return of the haematocrit to the patient's baseline. The duration of illness was taken as the number of days between the day of onset of illness and recovery.

## Statistical analysis

Assuming a response rate of 42.8% by the active drug rupatadine in reduction of fluid leakage measured by ultrasound scan, and with the standard treatment which would reduce fluid leakage by 23.5%, the number of participants per group that required to detect a difference in the two groups with a significance level of 5% and a power of 1 - β was calculated to be 140 in each arm. With a planned interim analysis, the significance value of less was set at 0.05 at the final analysis to indicate statistical significance. In accordance with the intention-to-treat principle, all patients as per the inclusion and exclusion criteria were included in the group to which they were randomly assigned. If and when a patient is unblinded that patient was not included in the analysis. Any data from a patient who was lost to follow up was included until that point.

As the study was prematurely terminated when 123 were recruited to the rupatadine arm and 126 to the placebo arm, resulting in a 0.05 type I error, the study was found to be underpowered at a post hoc power of 43.3% to evaluate the primary end point, which was the reduction in the proportion of patients developing DHF.

At the end of the trial, the primary endpoint was the reduction in the number of individuals who develop DHF. And the secondary efficacy endpoints included duration of reduction in stay at hospitals and DHF associated complications. Statistical analysis was performed using Graph PRISM version 8.3 and non-parametric statistical tests were used. Differences in the serial values of the platelet counts, and white cell counts in patients on the two arms of treatment were done using multiple unpaired non-parametric t tests. Corrections for multiple comparisons were done using Holm-Sidak method and the statistical significant value was set at 0.05 (alpha). Longitudinal analyses were undertaken using 2-way repeated measures ANOVA. De-identified patient data is available as S1_Data.

## Results

### Patient characteristics

Although we planned to recruit 140 patients into each study arm, only 123 patients were recruited to the rupatadine arm and 126 patients into the placebo arm and the trial had to be stopped prematurely, mid-February 2020, as the hospital was converted to a COVID-19 treatment hospital. Although 137 patients were initially recruited to the rupatadine arm and 134 to the placebo arm, 8 patients from the placebo arm and 14 patients in the rupatadine arm were lost to follow up after the first day and were not contactable. Since no information regarding them was available, these patients were excluded from the analysis. The number of patients recruited to each treatment arm and the number followed up is shown in Fig 1. The baseline characteristics of the study population is shown in Table 1. During the 26-month study period, DENV1, DENV2 and DENV3 were circulating but none of the patients were found to be infected with DENV4.

### Primary outcomes: Efficacy of rupatadine in reducing the incidence of DHF and associated complications

The clinical features of those who were on the treatment and the placebo arm are shown in Table 2. 80 (65%) of patients in the rupatadine arm and 88 (69.8%) of patients in the treatment

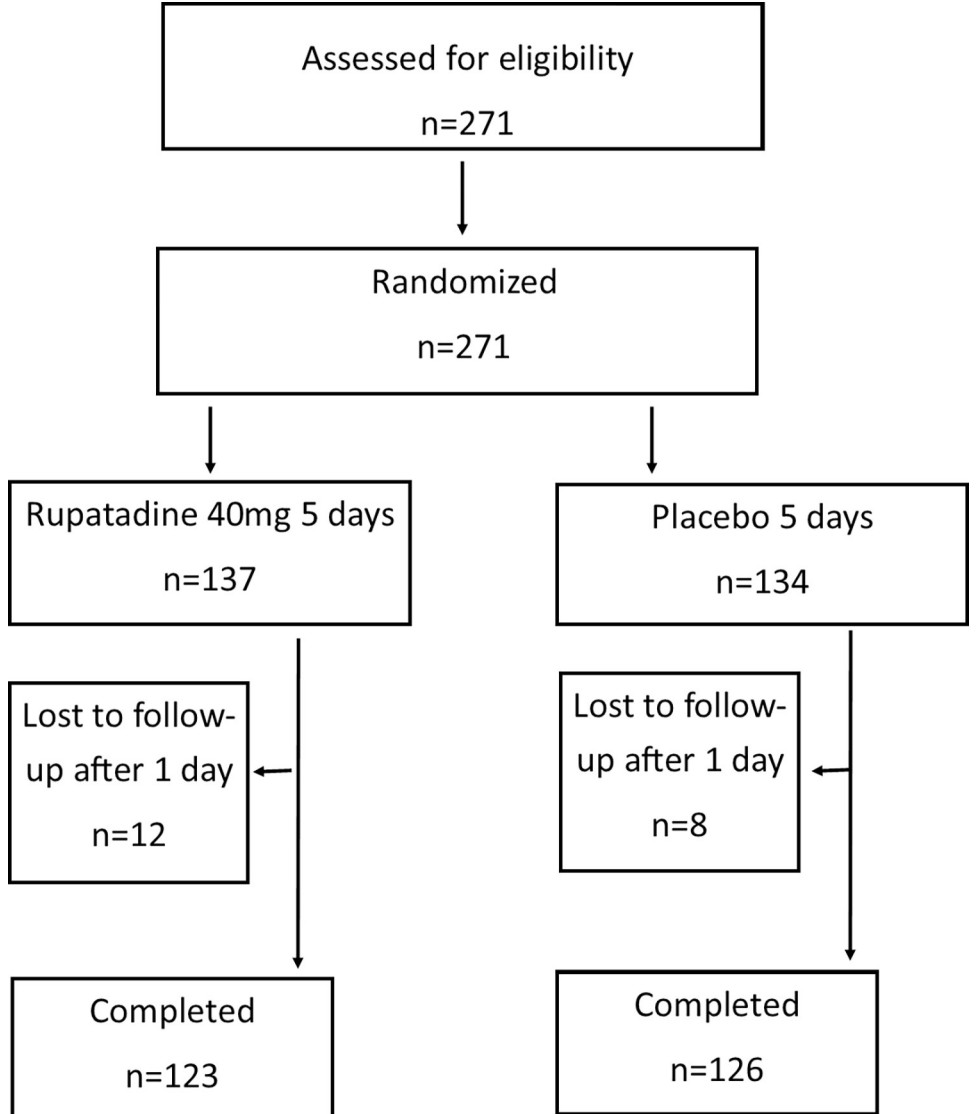

**Fig 1. Flow chart of the study showing recruitment to the two different arms of treatment.** 271 individuals were assessed for eligibility and randomised. 12 individuals in the rupatadine arm and 8 individuals in the placebo arm were lost to follow up.

arm were admitted to hospital. Of those who were admitted to hospital 12 (9.7%) in the treatment arm and 22 (17.5%) in the placebo arm developed DHF. Although these differences were not significant (p = 0.09), rupatadine was associated with a relative risk of DHF of 0.68 (95% of 0.41 to 1.04). As this was an outpatient trial very few patients developed bleeding manifestations (6 patients on rupatadine and 7 on the placebo) and shock (2 in the rupatadine arm and 3 in the placebo arm), and therefore, the numbers were inadequate for statistical significance. The bleeding manifestations that were seen included vaginal bleeding in 5/6 patients on rupatadine and 5/7 patients on the placebo.

We had previously reported that rupatadine 40mg was safe and well tolerated in patients with acute dengue [12]. There was no incidence of an increase in adverse reactions such as headache, vomiting and changes in haematological parameters in patients in the treatment

**Table 1. The baseline characteristics of those who were recruited to receive either 40mg of rupatadine daily for 5 days or the placebo for 5 days.**

| Clinical and Laboratory Characteristics | Rupatadine N = 123 | Placebo N = 126 | P value |
|---|---|---|---|
| Duration of illness at time of presentation (median, IQR) | 3 (2 to 3) | 3 (2 to 3) | 0.23 |
| Day of illness at time of recruitment<br>Day 1<br>Day 2<br>Day 3 | <br>4<br>35<br>84 | <br>9<br>40<br>77 | |
| Age (median, IQR) | 28 (21 to 38) | 28.5 (22 to 39) | 0.68 |
| Gender male: female | 48:77 | 56:71 | |
| Infecting DENV serotype<br>DENV1<br>DENV2<br>DENV3<br>DENV4<br>negative | <br>33<br>55<br>15<br>0<br>20 | <br>35<br>57<br>22<br>0<br>12 | |
| Viral loads at presentation (copies/ml) | 227,717 (9,242 to 5,484,384) | 280,992 (4,727 to 5,132,283) | 0.64 |
| Wellness score at time of recruitment (median, IQR) | 7 (5 to 8) | 7 (6 to 8) | 0.37 |
| WBC (median, IQR) | 4.37 (3.5 to 5.77) | 4.5 (3.4 to 6.1) | 0.62 |
| Platelet counts (median, IQR) | 174 (131 to 220.5) | 166 (130 to 213.5) | 0.37 |

arm. In fact, the proportion of individuals on rupatadine who developed vomiting (p = 0.04) and headache (p = 0.04) were significantly lower, which are clinical features seen in patients with acute dengue. Importantly, those who received rupatadine were significantly less likely to develop hepatic tenderness (p<0.0001) and a significant reduction in the duration of illness (p = 0.002). None of the patients on rupatadine reported any hypersensitivity reactions, irritability, psychomotor impairments, and there was no difference in the appetite in the treatment arm compared to the placebo arm. There was no difference in the white cell counts in those who were on the treatment arm compared to the placebo arm during any day of treatment.

**Table 2. Clinical and laboratory features in those who were given rupatadine 40mg for 5 days or a placebo for 5 days.**

| Clinical and Laboratory Characteristics | Rupatadine N = 123 | Placebo N = 126 | Relative risk (95% CI) | P value |
|---|---|---|---|---|
| Abdominal pain | 28 (22.8%) | 28 (22.2) | 1.01 (0.73 to 1.13) | >0.99 |
| Persistent vomiting | 9 (8.3%) | 20 (15.9%) | 0.59 (0.32 to 0.95) | **0.04** |
| Headache | 108 (87.8%) | 120 (95.2%) | 0.66 (0.52 to 0.97) | **0.04** |
| Diarrhoea | 35 (28.4%) | 37 (29.3%) | 0.97 (0.72 to 1.3) | 0.88 |
| Reduced appetite | 101 (82.1%) | 99 (78.6%) | 1.1 (0.82 to 1.6) | 0.52 |
| Hepatic tenderness | 6 (4.9%) | 12 (9.7%) | 0.37 (0.18 to 0.63) | **<0.0001** |
| Admission to hospital | 80 (65%) | 88 (69.8%) | 0.89 (0.69 to 1.12) | 0.49 |
| Development of DHF | 12 (9.7) | 22 (17.5) | 0.68 (0.41 to 1.04) | 0.09 |
| Ascites | 12 (9.7) | 22 (17.5) | 0.68 (0.41 to 1.04) | 0.09 |
| Pleural effusions | 3 (2.4%) | 2 (1.6%) | 1.2 (0.46 to 1.9) | 0.68 |
| Bleeding manifestations (excluding cutaneous bleeding) | 6 (4.8%) | 7 (5.47) | 0.93 (0.46 to 1.47) | >0.99 |
| Dengue shock | 2 (1.6%) | 3 (2.4) | 0.81 (0.24 to 1.6) | >0.99 |
| Dextran given | 3 (2.4%) | 5 (3.9) | 0.75 (0.27 to 1.4) | 0.72 |
| Normal saline boluses given | 3 (2.4%) | 5 (3.9) | 0.75 (0.27 to 1.4) | 0.72 |
| Blood given | 0 (0) | 2 (1.6) | 0.0 (0. to 1.13) | 0.49 |
| Platelet counts (nadir of thrombocytopenia)<br><20,000<br><50,000 | <br>13 (10.6%)<br>21 (17.1%) | <br>14 (10.9%)<br>39 (30.5%) | <br>1.0 (0.63 to 1.4)<br>0.64 (0.43 to 0.90) | <br>>0.99<br>**0.01** |
| Duration of fever (median, IQR) | 4 (3 to 5) | 4 (3 to 5) | | 0.93 |
| Duration of illness (median, IQR) | 5 (3 to 7) | 6 (4 to 7.25) | | **0.0002** |

The only serious adverse effect noticed in the patients was hospitalization, as some required admission as they either fulfilled the criteria of hospital admission according to the National dengue management guidelines or they wished to be admitted due to social circumstances. There was no difference in hospitalization rates between the treatment arm (p = 0.49) compared to the placebo arm.

### Effect on rupatadine on haematological parameters

In our preliminary study, we previously reported that those who were given rupatadine had a significant reduction of the extent of thrombocytopenia on day 7 of illness [12]. In this study the proportion of patients on rupatadine (16.7%) who had a reduction of platelets counts of <50,000 cells/mm$^3$ was significantly less (p = 0.01) than those who were on the placebo (30.5%) (Table 2), with a relative risk of 0.64 (95% CI, 0.43 to 0.9). As seen previously, the reduction in platelet counts were significantly less on day 7 of illness in those who received rupatadine compared to those who were on the placebo arm (Fig 2A). On day 9 of illness, 6

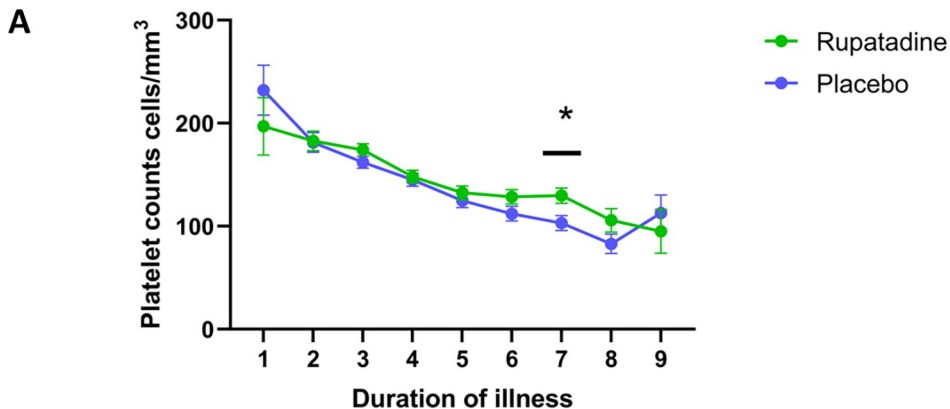

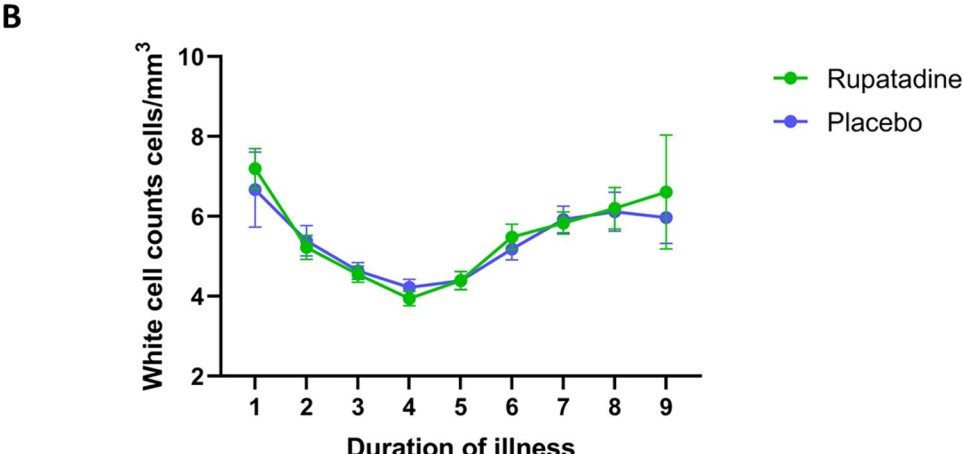

**Fig 2. The change in the platelet counts and white cell counts in patients with acute dengue infection.** Platelet counts were measured daily in patients with acute dengue infection who were on rupatadine 40mg daily (n = 123) or a placebo daily (n = 126). Data shown for day 9 includes only 6 patients on rupatadine and 7 patients on the placebo as many recovered and were not followed up until this time (A). Total white cell counts were also measured daily in patients with acute dengue infection who were on rupatadine 40mg daily (n = 123) or a placebo daily (n = 126) (B). Differences the two arms of treatment were done using multiple unpaired non-parametric t tests. The lines indicate the mean and the error bars the standard error of the mean. All tests were two-tailed.

patients on rupatadine and 7 patients on the placebo were still being followed up and therefore, were included in the analysis. Of the 6 patients on rupatadine 1/6 had a platelet count of <10,000 cells/mm$^3$ on day 9, and 2/6 patients had platelet counts between 50,000 to 100,000 cells/mm$^3$. In the placebo arm 4/7 patients had platelet counts between 50,000 to 100,000 cells/mm$^3$ on day 9 of illness. There was no difference in the total white cell counts in those who were given rupatadine compared to those who were on the control arm (Fig 2B).

## Discussion

In this study we assessed the efficacy of rupatadine 40mg for 5 days in reducing the incidence of DHF in patients with acute dengue infection. Although we aimed to recruit and follow up 280 patients (140 in each arm), the trial had to be prematurely terminated as the trial centre (National Institute of Infectious Diseases), was converted to a COVID-19 hospital in February 2020. Therefore, only 123 patients completed treatment in the rupatadine arm and 126 in the placebo arm. We found that 9.7% in the rupatadine arm developed DHF compared to 17.5% in the placebo arm, although this was not significant. Furthermore, they were significantly less likely to have reduction in platelet counts <50,000 cells/mm$^3$ and were significantly less likely to develop vomiting, headache and hepatic tenderness. However, the platelet data must be interpreted carefully as neither arm showed recovery of platelet counts by end of data collection.

Although the reasons for headache, vomiting and hepatic tenderness in dengue is not clear, it could be due to the high circulating levels of cytokines such as IL-6, TNFα, IL-1β and mast cell products, which are known to be elevated in patients with acute dengue [10,20,21]. Inflammation of the liver in dengue has shown to be multi-factorial including direct damage due to the virus, hypoxic damage due to shock and immune mediated inflammation [18]. As PAF has shown to induce production of these cytokines from monocytes and other immune cells [22], PAF receptor blockers such as rupatadine, may lead to a reduction of their production and therefore, reduce headache, vomiting and hepatic tenderness. Therefore, it would be important to evaluate the effects of rupatadine on serum cytokine levels and mast cell products in patients with acute dengue, in order to further understand possible mechanisms leading to the reduction of these clinical features. Interestingly, although those who were on rupatadine were less likely to have platelet counts <50,000 cells/mm$^3$, there was no difference in the bleeding manifestations, which was predominantly per vaginal bleeding in the treatment and placebo arm. Although many different mechanisms are responsible for bleeding in acute dengue [23], reduction of platelet counts alone, by rupatadine, did not seem to have any effect.

In our previous study, we showed that rupatadine also reduced the extent of the height of pleural effusions and rise in liver enzymes [12]. However, as this was an outpatient-based study, blood samples were only taken to assess changes in haematological parameters and therefore, changes in liver enzymes or the kinetics in changes in viral loads could not be assessed, although it would have been important to have measured them. Furthermore, only 3 individuals in the rupatadine arm and 2 in the placebo arm developed pleural effusions and therefore, effect of rupatadine on the extent of pleural effusions could not be assessed.

Based on our data, although rupatadine did appear to reduce the proportion of patients with acute dengue developing DHF, it is unlikely to be effective alone. Although significant increase in the ZO-1 expression and trans endothelial electrical resistance was seen in our previous in vitro studies, we used rupatadine at a concentration of 500ng/ml for in vitro studies and 0.8mg to 3mg/Kg for mouse studies [12]. In our clinical trial, the rupatadine dose used was 0.8mg/Kg in an adult who was 50Kg and therefore, the concentrations of rupatadine used was far less than the concentrations used in the in vitro and mouse studies. Therefore, in our

trial although rupatadine did appear to have a significant effect on reducing headache, vomiting and hepatic tenderness with and also a trend towards reducing vascular leak and reducing the extent of thrombocytopenia, it is unlikely that rupatadine is likely to be effective by itself, when given at 40mg daily doses.

In addition to PAF, many other mediators are responsible for the vascular leak in acute dengue [7]. Since rupatadine is an antihistamine and PAFR blocker, it is unlikely to influence preventing vascular leak that occurs due to other mediators that result in vascular leak such as IL-1β, TNF-α and dengue NS1 antigen [7,8,24]. We recently showed that leukotrienes and histamines could play a role in the vascular leak of acute dengue as urinary leukotriene (LT4) was found to be significantly higher and remained high in patients who developed DHF, while histamine was significantly higher in patients with acute dengue irrespective of disease severity [25]. 24 hours urinary histamine levels have shown to be high in patients with DHF by others too [26]. Histamine has shown to result in increased vascular permeability by changing the VE-Cadherin localization on endothelial cells and also causing changes in the actin cytoskeleton [27,28]. Therefore, it is possible that some of the effects of rupatadine could be due to histamine blockade in addition to blockage of PAF receptor. LTE4 has also shown to induce vascular leak, by causing contraction of endothelial cells, in a dose dependent manner [29]. Therefore, it would be important to study the efficacy of drugs that also inhibit the action of cysteinyl leukotrienes such as montelukast [30]. Indeed, montelukast has been safely used with many other antihistamines in patients with asthma, allergic rhinitis and chronic urticaria [30,31]. As both montelukast and rupatadine are widely used and relatively cheap, it would be important to explore if the combination of rupatadine along with montelukast could further reduce the incidence of DHF.

## Supporting information

**S1 Appendix. Detailed trial protocol.**
(DOCX)

**S1 Data. Raw data of patients given either rupatadine 40mg for 5 days or the placebo that was used for analysis and generation of tables and figures in this manuscript.**
(XLSX)

## Author Contributions

**Conceptualization:** Gathsaurie Neelika Malavige, Chandima Jeewandara, Ananda Wijewickrama, Chandanie Wanigatunga, Harsha Dissanayake, Shamini Prathapan, Graham S. Ogg.

**Data curation:** Dumni Gunasinghe, Sameera D. Mahapatuna, Chathurika Gangani, Vimalahan Vimalachandran, Geethal Jayarathna, Yashoda Perera, Samurdhi Wickramanayake, Pramodth Sahabandu.

**Formal analysis:** Gathsaurie Neelika Malavige, Harsha Dissanayake, Laksiri Gomes.

**Funding acquisition:** Gathsaurie Neelika Malavige, Graham S. Ogg.

**Investigation:** Dumni Gunasinghe, Laksiri Gomes.

**Methodology:** Laksiri Gomes.

**Project administration:** Gathsaurie Neelika Malavige, Chandima Jeewandara, Ananda Wijewickrama, Eranga Narangoda, Damayanthi Idampitiya.

**Resources:** Gathsaurie Neelika Malavige, Chandima Jeewandara, Ananda Wijewickrama, Eranga Narangoda, Graham S. Ogg.

**Supervision:** Gathsaurie Neelika Malavige, Chandima Jeewandara, Ananda Wijewickrama, Damayanthi Idampitiya.

**Writing – original draft:** Gathsaurie Neelika Malavige, Graham S. Ogg.

**Writing – review & editing:** Gathsaurie Neelika Malavige, Graham S. Ogg.

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
