## [Decision Letter · Decision Letter 0]

28 Jan 2022

Dear Professor Malavige,

Thank you very much for submitting your manuscript "Efficacy of rupatadine in reducing the incidence of dengue haemorrhagic fever in patients with acute dengue: a randomised, double blind, placebo-controlled trial" for consideration at PLOS Neglected Tropical Diseases. As with all papers reviewed by the journal, your manuscript was reviewed by members of the editorial board and by several independent reviewers. In light of the reviews (below this email), we would like to invite the resubmission of a significantly-revised version that takes into account the reviewers' comments. 

We cannot make any decision about publication until we have seen the revised manuscript and your response to the reviewers' comments. Your revised manuscript is also likely to be sent to reviewers for further evaluation.

Sincerely,

William B Messer

Associate Editor

Tereza Magalhaes

Deputy Editor

Reviewer's Responses to Questions

**Key Review Criteria Required for Acceptance?**

**Methods**

-Are the objectives of the study clearly articulated with a clear testable hypothesis stated?

-Is the study design appropriate to address the stated objectives?

-Is the population clearly described and appropriate for the hypothesis being tested?

-Is the sample size sufficient to ensure adequate power to address the hypothesis being tested?

-Were correct statistical analysis used to support conclusions?

-Are there concerns about ethical or regulatory requirements being met?

Reviewer #1: Malavige et al described a double blind RCT in patients with acute dengue in Sri Lanka in an outpatient setting comparing oral rupatadine ( an antihistamine known to have long acting dual histamine-1-receptor blocking activities and PAF receptor blocking activities for the treatment of allergic diseases) 40mg for 5 days with placebo in reducing the incidence of dengue haemorrhagic fever (DHF) in patients with acute dengue. Prev preclinical studies fromo the same group have demonstrated that PAF receptor blockade inhibited the effects of acute dengue sera on endothelial permeability and in reducing vascular leak in a dengue serotype 2 mouse model. However, an earlier phase 2 trial in 134 dengue patients ( 67 in the rupatadine group and 67 in placebo group) failed to show a benefit in the primary efficacy endpoint of ascites or pleural effusions when the drug was given within 5 days of symptoms. Post hoc analysis however revealed that in those who received Rupatadine within 3 days of symptoms, there was a trend towards more reduction in the fluid leakage, as assessed by the degree of ascites (mild or moderate). Based on their earlier findings, the authors then conducted a larger phase 2 double blind RCT, results of which are presented in this current report. 

The current trial involved a larger sample size of 249 dengue patients ( 123 received rupatadine and 126 received a placebo, once a day for 5 days) in an outpatient setting.

Overall, the study design, rationale and data are well described and clearly presented.

Reviewer #2: Most of the methods are well-described except for the sample size calculation. Please see summary and general comments for details.

**Results**

-Does the analysis presented match the analysis plan?

-Are the results clearly and completely presented?

-Are the figures (Tables, Images) of sufficient quality for clarity?

Reviewer #1: As above

Reviewer #2: Yes. There are a few areas that could be improved and further analysed as indicated in the summary and general comments section.

**Conclusions**

-Are the conclusions supported by the data presented?

-Are the limitations of analysis clearly described?

-Do the authors discuss how these data can be helpful to advance our understanding of the topic under study?

-Is public health relevance addressed?

Reviewer #1: The conclusions are generally sound although this reviewer has recommended the authors to tone down on some claims and their language used for clarity.

Reviewer #2: The conclusion should be revised as indicated below.

**Editorial and Data Presentation Modifications?**

Reviewer #1: (No Response)

Reviewer #2: Please see below.

**Summary and General Comments**

Reviewer #1: The manuscript is generally well written and data has been clearly presented.

My comments are as follow:

1. The abstract and summary are misleading in their claims. I would suggest the authors tone down the language used.

Line 35-36: Therefore, patients given rupatadine were 55% less likely to develop DHF than those on the placebo (relative risk 0.68, 95% CI 0.41 to 1.08). This statement is not accurate as the statistical analysis did not reach significance.

Lines 63-65: Rupatadine was previously shown to reduce endothelial dysfunction in vitro, reduced vascular leak in dengue mouse models and to reduce the extent of pleural effusions and thrombocytopenia in patients with acute dengue.

I would suggest that the authors clarify that the reduction in vascular leak observed in humans only showed a trend and was a post hoc analysis in the prev study published in Sci Report.

Lines 70-71: Same comments as above.

2. it would be important to state clearly under the statistical analysis section whether the 140 per arm in the original sample size calculation included anticipated drop-outs. And with the final sample size of 123/126, what is the power of this trial to detect a difference in the reduction of fluid leakage in the 2 arms. If the current trial size is adequately powered to address the primary endpoint, the authors should not be advocating for yet another larger trial of rupatadine monotherapy in their summary and conclusion of the discussion. A larger trial is unlikely to be useful at the expense of cost and time if the current trial size is powered adequately.

3. It is interesting that patients in the rupatadine arm experienced less hepatic tenderness, vomiting and headache. These findings should be discussed further as to the possible/postulated mechanisms of actions involved. Liver inflammation is one of the causes of hepatic tenderness in acute dengue. How did the liver enzymes compare in the 2 arms at baseline ?

4. In the discussion lines 314-316: Therefore, although we only 123 patients completed treatment in the rupatadine arm and 126 in the placebo arm, we found that those who were given rupatadine were twice as less likely (9.7%

compared to 17.5%) to develop DHF compared to those on the placebo arm.

This statement is misleading as the result was not statistically significant.

It is also noteworthy that the rupatadine arm had fewer patients experiencing platelet counts of less than 50,000 compared to placebo although this did not translate to fewer bleeding events overall.

Reviewer #2: This manuscript by Malavige and colleagues report the findings of a phase 2 clinical trial into the rupatadine as a therapy to prevent dengue hemorrhagic fever (DHF). The authors have previously found that rupatadine reduced endothelial cell dysfunction and hence vascular leakage in experimental models. In a previously published proof-of-concept trial, they also showed that rupatadine reduced the rate of pleural effusion although treatment did not reduce the rate of patients that eventually developed DHF compared to placebo. In this trial, the goal was to test the hypothesis that rupatadine, when given early following illness onset, would reduce the extent of vascular leakage and hence the rate of DHF in dengue patients. The authors randomised 123 patients into receiving rupatadine and 126 patients into receiving placebo. The primary endpoint, reduction in relative risk of DHF, was not statistically significant. There were secondary endpoints that were statistically significant. The authors surmised that the primary endpoint was compromised by the failure to recruit 140 cases per arm due to the reorganization of the public health resource allocation to manage the covid-19 outbreak in Sri Lanka. They thus concluded that the findings support a larger trial to demonstrate the efficacy of rupatadine in preventing DHF.

This is a well written manuscript that reports an important clinical trial. Several areas, however, need attention. These are:

1. The description of the a priori sample size calculation in the main manuscript is inconsistent with the supplementary information. The main text omitted the mention of n=140 was derived to cater for the anticipated 10% dropout rate. The number of patients that need to be included for a statistically meaningful analysis, using the equation shown in the supplementary information, should thus be reported, in addition to the final estimate of n=140.

2. Following from the above comment, the inclusion of n=123 and n=126 in the active and placebo arms, respectively, is within the range of the a priori calculated sample size. The conclusion in lines 43-45 is thus not convincing. What is clear from this and the previous smaller clinical trial carried out by the authors is that data from the experimental models have exaggerated the therapeutic effect of rupatadine on vascular leakage. The assumed response rate is thus overly optimistic. The authors should thus consider a more circumspect discussion on the trial findings and what the data tells on the a priori assumptions. They should also discuss whether a response rate that is more reasonable and lower than presently assumed would still be clinically useful. A recommendation for future trials should only be made after a more thorough and circumspective discussion on the findings.

3. The authors should also consider taking the opportunity to discuss biological insights that can be gleaned from their trials. Clearly, rupatadine does have some effect on reducing liver dysfunction in dengue patients, as well as reduce the rate/intensity of some of the classical dengue symptoms. What can the dengue field glean about dengue pathogenesis from these different outcomes?

4. Lines 110-118. This justification for anti-DHF therapy is unnecessary. Dengue is both a medical and a global health problem. The need for such therapy is independent of the problems caused by covid-19.

5. Line 150. Please specify if the NS1 antigen test used was point-of-care test or ELISA.

6. Table 2. It is unclear if the platelet counts shown in the table represent the nadir of thrombocytopenia. Please clarify.

7. Figure 2A is somewhat concerning. The platelet count in the treated arm continued on the downward trend even at the end of the monitoring period whereas those in the placebo arm has shown an upswing. Could rupatadine be delaying the platelet nadir or even prolong thrombocytopenia?

PLOS authors have the option to publish the peer review history of their article (what does this mean?). If published, this will include your full peer review and any attached files.

Reviewer #1: No

Reviewer #2: No
---

## [Decision Letter · Decision Letter 1]

29 Mar 2022

Dear Professor Malavige,

Thank you very much for submitting your manuscript "Efficacy of rupatadine in reducing the incidence of dengue haemorrhagic fever in patients with acute dengue: a randomised, double blind, placebo-controlled trial" for consideration at PLOS Neglected Tropical Diseases. As with all papers reviewed by the journal, your manuscript was reviewed by members of the editorial board and by several independent reviewers. In light of the reviews (below this email), we would like to invite the resubmission of a significantly-revised version that takes into account the reviewers' comments. 

In addition to the edits suggested by the reviewer, we would ask that you add as supplementary data or otherwise make available the coded (de-identified) subject level trial data used to generate the tables and figures.

We cannot make any decision about publication until we have seen the revised manuscript and your response to the reviewers' comments. Your revised manuscript is also likely to be sent to reviewers for further evaluation.

Sincerely,

William B Messer

Associate Editor

Tereza Magalhaes

Deputy Editor

Reviewer's Responses to Questions

**Key Review Criteria Required for Acceptance?**

**Methods**

-Are the objectives of the study clearly articulated with a clear testable hypothesis stated?

-Is the study design appropriate to address the stated objectives?

-Is the population clearly described and appropriate for the hypothesis being tested?

-Is the sample size sufficient to ensure adequate power to address the hypothesis being tested?

-Were correct statistical analysis used to support conclusions?

-Are there concerns about ethical or regulatory requirements being met?

Reviewer #2: The sample size calculation needs to be reported in greater detail. Please see comments below.

**Results**

-Does the analysis presented match the analysis plan?

-Are the results clearly and completely presented?

-Are the figures (Tables, Images) of sufficient quality for clarity?

Reviewer #2: The issue with the platelet count on day 9 was incorrectly dealt with. Please see comments below for more details.

**Conclusions**

-Are the conclusions supported by the data presented?

-Are the limitations of analysis clearly described?

-Do the authors discuss how these data can be helpful to advance our understanding of the topic under study?

-Is public health relevance addressed?

Reviewer #2: Depending on the sample size calculation and platelet count analysis, the conclusion may need to be phrased differently.

**Editorial and Data Presentation Modifications?**

Reviewer #2: Minor comments:

1. Lines 35-37 is repetitive. Details on relative risk etc can be incorporated into the preceding sentence and the sentence in lines 35-37 can be deleted.

2. Line 65. Post-ad hoc should be changed to post hoc.

3. Lines 313-316. Please revise this sentence.

**Summary and General Comments**

Reviewer #2: I thank the authors for addressing some of the concerns raised on their original submission. However, several issues remain to be fully resolved. These are:

1. The authors missed out on responding to my original comment #1. The number of patients that need to be included for a statistically meaningful analysis, based on the equation shown in the supplementary information and before accounting for anticipated dropout rates need to be reported. Without this important detail, the amended lines 215-217 remains unsubstantiated.

2. The removal of day 9 platelet count data from Figure 2A is incorrect. I fully understand the authors’ explanation, but removal of that data is not the way to solution. The authors should instead reinstate this data but provide an explanation in the figure legend on how the day 9 platelet count data should be interpreted.

3. The amended Figure 2A without a corresponding change in Table 2 raises another concern: Was day 9 platelet count included in the analysis on the nadir of platelet count? Clearly, for some patients in the active arm, the platelet count may still be on the declining trend at day 9. The possibility that rupatadine treatment delays platelet recovery must be considered and instead of being brushed aside.

PLOS authors have the option to publish the peer review history of their article (what does this mean?). If published, this will include your full peer review and any attached files.

Reviewer #2: No
---

## [Editor Report · Decision Letter 2]

7 May 2022

Dear Professor Malavige,

Thank you very much for submitting your manuscript "Efficacy of rupatadine in reducing the incidence of dengue haemorrhagic fever in patients with acute dengue: a randomised, double blind, placebo-controlled trial" for consideration at PLOS Neglected Tropical Diseases. As with all papers reviewed by the journal, your manuscript was reviewed by members of the editorial board and by several independent reviewers. The reviewers appreciated the attention to an important topic. Based on the reviews, we are likely to accept this manuscript for publication, providing that you modify the manuscript according to the review recommendations. 

Reviewer 2's concern that the data in figure 2A suggest that rupatadine may delay recovery of platelet counts has not been fully addressed. As I read the reviewer's comments, it is clear that the possibility that rupatadine could delay platelet count recovery needs to be mentioned in the manuscript. The fact that platelet counts were not tracked to recovery means that any claims of rupatadine treatment diminishes thrombocytopenia are not fully substantiated by the data. Please add the observation that neither arm demonstrated a recovery in platelet counts by day 9 to the results section and then add to the discussion an acknowledgement that platelet data must be interpreted carefully as neither arm showed recovery of platelet count by end of data collection.

Additional minor comments

line 190 - Ultra sound should be one work and not capitalized

Lines 353-360 can be deleted as they are redundant with early text.

Sincerely,

William B Messer

Associate Editor

Tereza Magalhaes

Deputy Editor

Reviewer 2's concern that the data in figure 2A suggest that rupatadine may delay recovery of platelet counts has not been fully addressed. As I read the reviewer's comments, it is clear that the possibility that rupatadine could delay platelet count recovery needs to be mentioned in the manuscript. The fact that platelet counts were not tracked to recovery means that any claims of rupatadine treatment diminishes thrombocytopenia are not fully substantiated by the data. Please add the observation that neither arm demonstrated a recovery in platelet counts by day 9 to the results section and then add to the discussion an acknowledgement that platelet data must be interpreted carefully as neither arm showed recovery of platelet count by end of data collection.

Additional minor comments

line 190 - Ultra sound should be one work and not capitalized

Lines 353-360 can be deleted as they are redundant with early text.

Figure Files:

Data Requirements:

Reproducibility:

References

---

## [Editor Report · Decision Letter 3]

16 May 2022

Dear Professor Malavige,

We are pleased to inform you that your manuscript 'Efficacy of rupatadine in reducing the incidence of dengue haemorrhagic fever in patients with acute dengue: a randomised, double blind, placebo-controlled trial' has been provisionally accepted for publication in PLOS Neglected Tropical Diseases.

Best regards,

William B Messer

Associate Editor

Tereza Magalhaes

Deputy Editor

---

## [Editor Report · Acceptance letter]

27 May 2022

Dear Professor Malavige,

We are delighted to inform you that your manuscript, "Efficacy of rupatadine in reducing the incidence of dengue haemorrhagic fever in patients with acute dengue: a randomised, double blind, placebo-controlled trial," has been formally accepted for publication in PLOS Neglected Tropical Diseases.

Best regards,

Shaden Kamhawi

co-Editor-in-Chief

Paul Brindley

co-Editor-in-Chief
